# Facebook addiction and its related factors among medical students; a cross- sectional study in Bangladesh

**Md Rizwanul Karim**[1]*, **Md. Jawadul Haque**[2], **Shahnaz Akhter**[3], **Helal Uddin Ahmed**[4]

**1** Department of Community Medicine, Patuakhali Medical College (PKMC), Patuakhali, Bangladesh,
**2** Department of Community Medicine, Rajshahi Medical College, Rajshahi, Bangladesh, **3** Department of Gynae and Obstetrics, Combined Military Hospital, Jalalabad, Sylhet Cantonment, Sylhet, Bangladesh,
**4** Department of Child Adolescent & Family Psychiatry, National Institute of Mental Health (NIMH), Shyamoli, Dhaka, Bangladesh

* Shameem.m25@gmail.com

**Data Availability Statement:** https://figshare.com/articles/dataset/Facebook_addiction_and_its_related_factors_in_medical_students_Bangladesh_context/20377236.

## Abstract

There is mounting evidence that Facebook Addiction is associated with poor mental health, physical symptoms, social dysfunction, and despair among the adolescent and youth population. The current study set out to identify the prevalence of Facebook Addiction among Bangladeshi medical students as well as its influencing factors. This cross-sectional survey was conducted on 720 randomly selected medical students from eight public medical colleges from January to June 2022. Data were obtained using a semi- structured, self-reported questionnaire and analyzed using the SPSS v.23 programs. The Bergen Facebook Addiction Scale BFAS was used to assess Facebook Addiction, while the Generalized Anxiety Disorder GAD-7, Patient Health Questionnaire PHQ-9, Perceived Stress Scale PSS10, Chen Internet Addiction Scale CIAS, and Pittsburg Sleep Quality Index PSQI was used to assess anxiety, depression, perceived stress, internet addiction, and sleep quality. Binary logistic regression was used to evaluate the influence of several demographic, psychological, and behavioral characteristics on the likelihood of respondents being addicted to Facebook. Facebook Addiction was observed in 29.4% of medical students. According to data, 63.7% of medical students reported mild to severe anxiety, 29.3% moderate to severe depression, and 84.9% moderate to high perceived stress. Lack of personal income [OR with (95% CI), 1.82 (1.13, 2.96)], poor academic performance [2.46 (1.45, 4.15)], moderate anxiety [2.45 (1.22, 4.92)], moderate perceived stress [5.87 (1.92, 17.95)], and moderately severe depression [2.62 (.97, 7.08)] were all found to play a significant role in the development of Facebook Addiction. However, living with parents [OR with (95% CI), .37 (.14, .95)] and positive family relationships [.40 (.18, .87)] reduces the likelihood of becoming addicted to Facebook. An integrated participative Behavioral and psychological intervention should be devised to reduce the risks of Facebook addiction in medical students while also improving their mental health-related quality of life.

**Funding:** This research project was funded by Bangladesh Medical Research Council BMRC (BMRC/HPNSP-Research 2021 -2022; I 471). The sponsor helped by paying the authors honoraria, printing the questionnaire, and purchasing stationery, but they had no further involvement in the study's planning, data collecting, analysis, publication decision, or manuscript preparation.

**Competing interests:** The authors have declared that no competing interests exist.

## Introduction

Facebook, the most popular Social Networking Service SNS in the world, was initially only available to Harvard students when it launched on February 4, 2004. In July 2021, there were 2.89 billion monthly active users and 1.78 billion daily active users on Facebook [1]. Unlike other social networking sites, Facebook (FB) enables people to share and connect with people all over the world using their everyday communication devices. Some of the reasons and motivations for Facebook's popularity include the ability to easily access updated features that facilitate communication with friends, meeting others based on shared interests, sharing pictures and videos, blogging, dating, and even gaming [1–4]. The most popular reasons for using FB are "communication-related", such as social relationship management and companionship-seeking [5,6]. Facebook has such a big impact on a person's daily activities and social interactions that using the site excessively is considered a sign of Facebook addiction [1], which is being studied extensively around the world [1,2,7]. Facebook users may lose control of their usage and show a psychological need to remain online, update their profile, and comment on other users' updates. Due to their alleged "Fear of Missing Out," or concern that they will lose friends and followers if they are not active on the social media platform, they become uncontrollable on Facebook [8,9].

The risk of Facebook addiction FA was observed to be around 40% among Bangladeshi students [7]. Being single, not exercising, sleeping irregularly, using Facebook excessively, and having depressive symptoms have all been linked to Facebook addiction [7]. Previous research has shown a connection between internet addiction and depression in adults [10], adolescents [11–13], and college students [14]. Three-fourths of the articles in a systematic review of comorbid psychopathology in pathological Internet use found a strong association between pathological Internet use and depression [15]. Depression showed the strongest association with pathological Internet use when compared to other mental comorbidities [14]. A meta-analysis of five trials found that the prevalence of depression was consistently higher in the Internet Addiction (IA) group than in the control group [16].

According to recent research, people who frequently use Facebook heavily to relieve daily stress and find relief and social support are more likely to develop FA [17]. People with more severe depression symptoms have many social interaction options on Facebook. They receive positive social feedback from their interaction partners that they frequently miss offline, such as "Likes" and encouraging comments. Such uplifting encounters might raise the possibility that depressed people in particular frequently seek solace online in the form of positive reinforcement on Facebook. However, this might make them more likely to get FA [2,18]. Therefore, it is reasonable to assume that depression symptoms moderate the relationship between daily stress and FA based on prior research [19]. Although the study in Vietnam showed that stressed people are more likely to be addicted to Facebook, which increases the risk of sleep disturbances [20].

Studies in Turkey and Pakistan showed, that weekly time commitment, social motivations, severe depression, anxiety, and insomnia were all positively associated with Facebook addiction [21,22], but no such relationship was found between Facebook addiction and academic performance or loneliness [22]. Furthermore, excessive Facebook use can cause societal, familial, and romantic relationship conflicts, disagreements, discussions, and jealousy, as well as the opposite [23–25]. In Bangladeshi studies, Facebook Addiction was predicted to be influenced by several risk factors, including relationship failure, a history of domestic abuse, stressful life events, lack of concentration in daily tasks/studies, sleep disturbances from using Facebook more at night, and depressive symptoms [26,27]. In addition, some of them had additional health problems, such as back and neck pain, headaches, declining eyesight, etc [27].

The majority of studies on Facebook and/or Internet overuse among Bangladeshi students (college, university) were specific institution based and conducted using a nonprobability sampling technique, which limited their generalizability. The current study's objectives are to determine the prevalence of Facebook addiction in a random sample of Bangladeshi medical students as well as to assess the social and psychological factors that are associated with Facebook addiction.

## Material and methods

### Study design

From January to June 2022, a representative sample of students from eight public medical colleges participated in this cross-sectional survey.

### Sampling and participants

The sample size required to estimate the true prevalence of Facebook Addiction was computed using Epitools, assuming a 95 percent confidence interval and a 5 percent sampling error. The estimated sample size was 621 based on a 39.7% prevalence of Facebook addiction [7], a sensitivity of 73 percent, and a specificity of 99 percent for the Bergen Facebook Addiction Scale polythetic scheme [28]. We planned to enroll 720 medical students using two-step stratified random sampling and correcting for 10% non-response and 5% missing value using the estimated sample. At first, eight of Bangladesh's 37 public medical colleges were chosen at random, each with around 250 to 1250 regular students. Following that, 90 students from each medical college were picked at random, with 18 students enrolling from each academic year (first to the fifth year) [18*5*8 = 720] [Fig 1].

### Ethical approval and consent to participate

The Bangladesh Medical Research Council (BMRC) Ethical Review Committee gave its approval before the study began (BMRC/HPNSP-Research IRB 2021–2022 I 471). All ethical issues related to this study were carefully addressed in accordance with the Helsinki Declaration. Before we began collecting data, we briefly explained the study's goals and objectives to the respondents. We then obtained their written informed consent using a separate consent form attached to the main questionnaire. Respondents were given the assurance that any information they provided, such as their names or any other information that could be used to identify them, would be kept confidential and anonymous by substituting codes for participant identifiers and storing data in a locked cabinet, and that the anonymized data and or study results would only be disseminated and published in the public interest.

### Data collection

Before the commencement of data collection, a letter detailing the study's goals was issued to the administration of the individual medical colleges. The research team, which consisted of the lead author, co-authors, supervisors, and data collectors, organized a discussion session on the study procedure (Aim of the Study and Instruction Manual for the Questionnaire) for each year's students with the assistance of the Community Medicine Department of the respective medical college. The explanation and discussion session lasted around 1.30 hours (one hour for presentation and thirty minutes for questions and answers).

The survey questionnaire was created in English before being translated and pretested in Bengali. The Bengali version of the questionnaire was translated and reverse-translated by two separate multilingual translators to guarantee consistency and avoid response bias. The questionnaire was presented in Bengali to the selected respondents, who were instructed to complete it in the classroom. The self-reported survey took them about 25 minutes to complete.

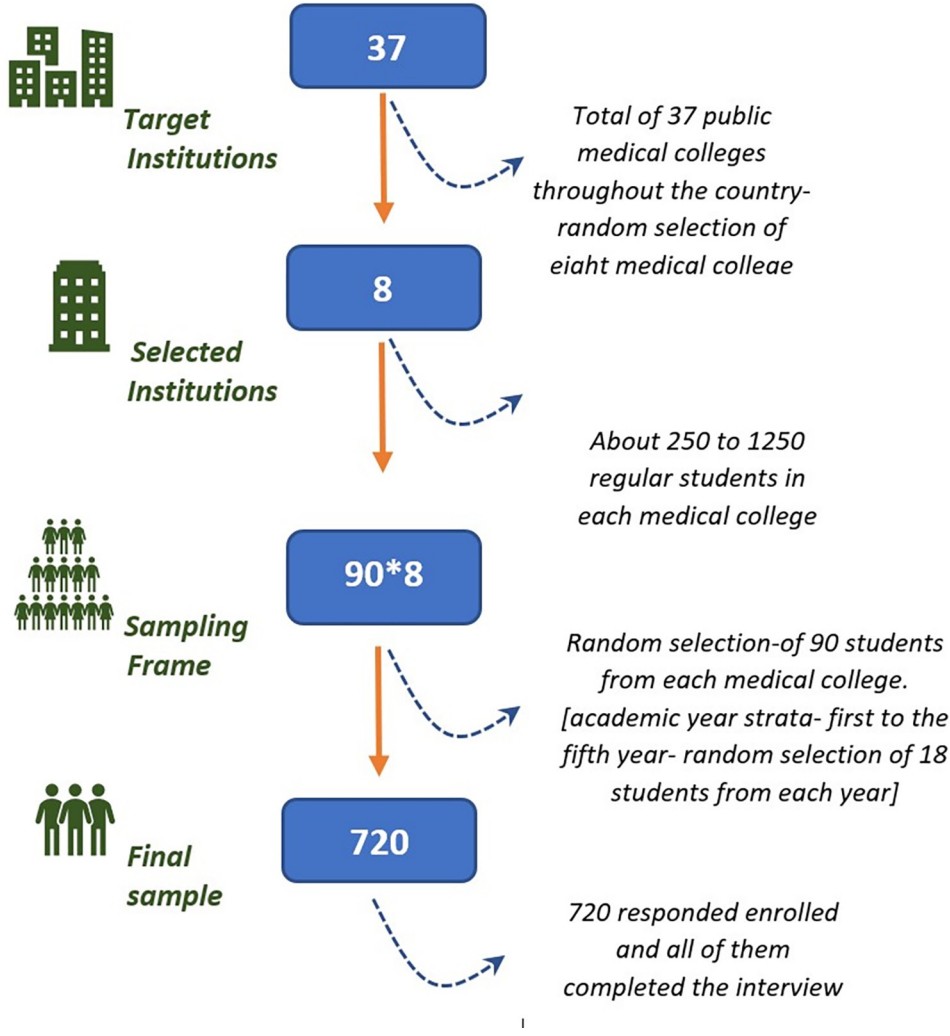

**Fig 1. Selection of participants.**

Following a review of the data for irregularities and missing information, it was observed that all 720 respondents had completed the whole survey. For further editing & analysis, the survey data was entered into the SPSS v-23 program.

## Measures

The questionnaire included 132 questions in total and was divided into eight sub-sections, as shown below. In addition to sociodemographic and Facebook use-related question items, the Bergen Facebook Addiction Scale was used to assess Facebook addiction, while the Generalized Anxiety Disorder GAD-7, Patient Health Questionnaire PHQ-9, Perceived Stress Scale PSS10, Chen Internet Addiction Scale CIAS, and Pittsburg Sleep Quality Index PSQI was used to assess anxiety, depression, perceived stress, internet addiction, and sleep quality.

## Socio-demographic variables

This section included several questions on socio-demographic variables, including age, gender, permanent residence, relationship status, Parental education, average monthly income, type and place of residence, religion, etc.

## Facebook uses related variables

The variables in this section were: a device used, type of network, time since they started using Facebook (FB), the primary purpose of Facebook use, average daily use, average monthly expenditure for Facebook use, total FB IDs, fake IDs, expenditure for FB use, number of friends on FB, followers in FB, the average number of posts in FB per day, average comment and reaction per FB post, self-measures to quit using the excessive use of FB. The three self-reported alternatives were utilized to analyze the following factors: financial status, academic performance, connections with friends and family, and romantic relationships with Facebook usage (average, negative, positive).

## Internet addiction

The CIAS is a self-reported four-point, a 26-item scale that examines five aspects of Internet-related behavioral traits, including compulsive usage, withdrawal, tolerance, interpersonal interaction problems, and health/time management concerns. The entire score range of the Chen Internet Addiction Scale is 26 to 84, with higher CIAS scores indicating more serious Internet Addiction [29]. Internal reliability for the scale and its subscales varied from 0.79 to 0.93 in the original research. According to Ko et al., the diagnostic cut-off point (63/64) provided the greatest diagnostic accuracy, Cohen Kappa, and DOR, and using this point, more than 80% of cases can be correctly classified [30]. Research including Iranian medical students showed the Chen Internet Addiction Scale (CIAS) to have acceptable levels of validity [(r = 0.85, (P = 0.001)] and reliability [Cronbach's alpha, (= 0.93)] [31].

## Facebook addiction

The Bergen Facebook Addiction Scale has six items with five potential responses on a Likert scale: 1 for very seldom, 2 for rarely, 3 for sometimes, 4 for often, and 5 for very often. This scale assesses the salience, tolerance, mood modification, conflict, relapse, and withdrawal experiences related to Facebook usage for the past year [32]. According to the authors, the BFAS can be used in both clinical and epidemiological situations, and they argue that a more liberal strategy based on a polythetic scoring system (e.g., scoring three or higher on at least four of the six items) is preferable to a more conservative strategy based on a monothetic scoring key (e.g., scoring 3 or above on all six items) [32]. A prior study showed the Bangla-translated BFAS had a high level of validity and reliability, and it could be used to test Facebook addiction among Bangladeshi university students [33]. However, the Confirmatory Factor Analysis CFA of the data from the current study further supports the usage of BFAS among medical students. The strongest item-total correlation (1 = .494, 4 = .594, 9 = .548, 10 = .663, 13 = .642, 16 = .606)) determined the retention of one out of three items from each sub-domain (salience, tolerance, mood modification, relapse, withdrawal, conflict) of BFAS, resulting in a suggested six-item (item 1, 4, 9, 10, 13, 16) single factor BFAS model [32]. [Table 1] The proposed CFA model showed satisfactory goodness of fit indices and fit metrics [Table 2] as well as acceptable factor loadings of the scale items (> .55) [Fig 2]. In addition, Cronbach's alpha, McDonald's ω, Guttman's λ2 for the six item single factor Bergen Facebook Addiction Scale were.78,.78, and.79, respectively, showing the scale's acceptable to good reliability.

## Perceived stress

The PSS-10 assesses how much of one's life is unmanageable, unexpected, and overwhelming. Participants are asked to reply to each item on a 5-point Likert scale ranging from 0 (never) to 4 (very often), indicating how frequently they had felt or thought a specific way in the previous

**Table 1. The Bergen Facebook Addiction Scale: Items and inter-correlations of ratings.**

| Item-Total Statistics | |
| --- | --- |
| **How often during the last year have you. . .?** | |
| | **Item-Total Correlation** |
| **Salience** | |
| **1. Spent a lot of time thinking about Facebook or planned use of Facebook.** | **.494***  |
| 2. Thought about how you could free more time to spend on Facebook. | .413 |
| 3. Thought a lot about what has happened on Facebook recently | .491 |
| **Tolerance** | |
| **4. Spent more time on Facebook than initially intended** | **.594***  |
| 5. Felt an urge to use Facebook more and more. | .530 |
| 6. Felt that you had to use Facebook more and more to get the same pleasure from it | .574 |
| **Mood modification** | |
| 7. Used Facebook to forget about personal problems. | .500 |
| 8. Used Facebook to reduce feelings of guilt, anxiety, helplessness, and depression. | .525 |
| **9. Used Facebook to reduce restlessness.** | **.548***  |
| **Relapse** | |
| **10. Experienced that, others have told you to reduce your use of Facebook but not listened to them.** | **.663***  |
| 11. Tried to cut down on the use of Facebook without success. | .618 |
| 12. Decided to use Facebook less frequently, but not managed to do so. | .651 |
| **Withdrawal** | |
| **13. Become restless or troubled if you have been prohibited from using Facebook.** | **.642***  |
| 14. Become irritable if you have been prohibited from using Facebook. | .571 |
| 15. Felt bad if you, for different reasons, could not log on to Facebook for some time. | .548 |
| **Conflict** | |
| **16. Used Facebook so much that it has harmed your job/studies.** | **.606***  |
| 17. Given less priority to hobbies, leisure activities, and exercise because of Facebook. | .529 |
| 18. Ignored your partner, family members, or friends because of Facebook. | .515 |

*Six items were retained in the final model/scale, one from each subdomain, according to the highest item-total correlation. [BFAS1, BFAS4, BFAS9, BFAS10, BFAS13, BFAS16].

month [34]. The scale scores range from 0 to 40, with the higher composite value indicating more stress [34]. The original English 10-item version of PSS by Cohen & Williamson was translated into Bangla by different researchers [35–37]. Islam et al observed a significant correlation r = .90, p < .01 between the PSS-10-B (Translated and adapted by Fahim et al) with the original English version of PSS-10 [36,37]. The test-retest reliability of the Bangla PSS-10 scale was found to be outstanding over two weeks, r = .94, p.01, indicating that the Bangla PSS-10 scale may be used to assess felt stress among Bangladeshi people [35].

## Depression

The Patient Health Questionnaire (PHQ-9) is a nine-item self-administered questionnaire designed to measure the presence and severity of depressive symptoms in the last two weeks. The overall PHQ-9 score ranges from 0 to 27 since each of the nine items can be graded from 0 ("not at all") to 3 ("nearly every day"). Simple cut-points of 5, 10, 15, and 20 successively signify mild, moderate, moderately severe, and severe depression. If only one screening cut-point is employed, data suggests that a PHQ-9 score of 10 or higher has an 88 percent sensitivity, an 88 percent specificity, and a positive probability ratio of 7.1 [38]. Cronbach's alpha for the

**Table 2. Fit indices and fit metrics for the single factor BFAS containing proposed factor items.**

| Fit indices | | Other fit measures | |
| --- | --- | --- | --- |
| **Index** | **Value** | **Metric** | **Value** |
| **Comparative Fit Index (CFI)** | **0.98** | **Root mean square error of approximation (RMSEA)** | **0.06** |
| **Tucker-Lewis Index (TLI)** | **0.96** | RMSEA 90% CI lower bound | 0.04 |
| Bentler-Bonett Non-normed Fit Index (NNFI) | 0.96 | RMSEA 90% CI upper bound | 0.08 |
| Bentler-Bonett Normed Fit Index (NFI) | 0.97 | RMSEA p-value | 0.27 |
| **Parsimony Normed Fit Index (PNFI)** | **0.58** | **Standardized root mean square residual (SRMR)** | **0.03** |
| Bollen's Relative Fit Index (RFI) | 0.95 | **Goodness of fit index (GFI)** | **0.99** |
| **Bollen's Incremental Fit Index (IFI)** | **0.98** | McDonald's fit index (MFI) | 0.99 |
| Relative Noncentrality Index (RNI) | 0.98 | Expected cross-validation index (ECVI) | 0.08 |

Bengali translation of PHQ9 was 0.837, gender differences were 0.839 for males and 0.841 for females, the Spearman-Brown Coefficient was 0.855, and the Guttman split-half Coefficient was 0.848, all of which support the scale's usage in the current study [39].

## Anxiety

GAD-7, a self-reported seven-item measure, is used to screen for generalized anxiety disorder. The response choices for each item range from 0 to 3 on a 4-point Likert scale (0 = not at all, 1 = a few days, 2 = more than half the days, and 3 = virtually every day). When the scores of all seven items are summed up, the GAD-7 total score has a range of 0 to 21. Based on receiver operating characteristics analysis for the GAD-7, various validation studies have established

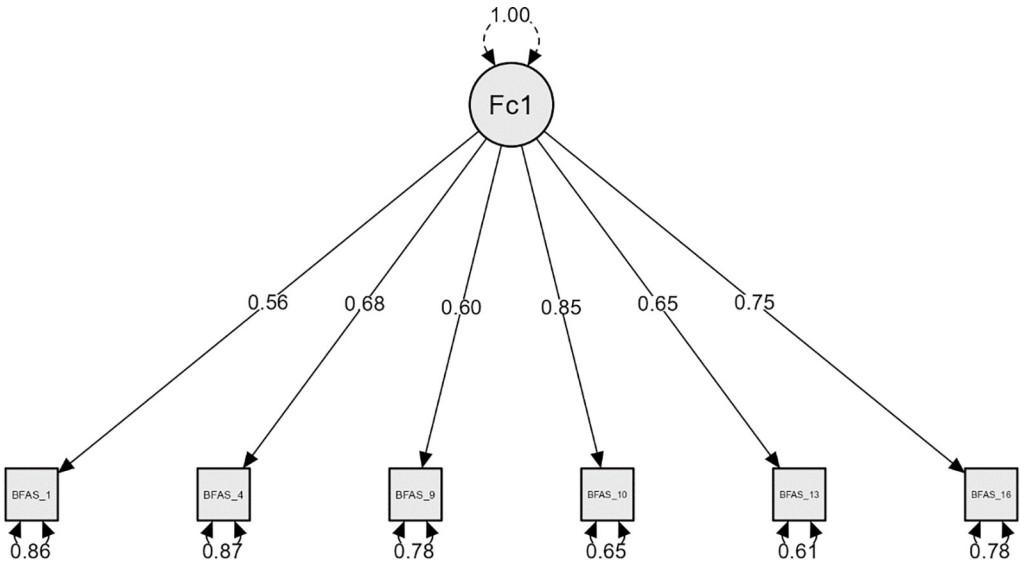

**Fig 2. Model plot: BFAS single factor loadings.**

cut-points of = >5, = >10, and = >15, which represent mild, moderate, and severe anxiety levels, respectively [40]. Researchers in Bangladesh reported that GAD-7 has high internal consistency (Cronbach's = 0.87) [41] and also showed good convergent validity with two other measures, PHQ-9 and PHQ- ADS [42].

## Sleep quality

The PSQI is a parameter-based questionnaire that depends on self-reported responses [43], and it is one of the most extensively used methods for measuring sleep quality. The PSQI consists of 19 questions divided into seven sleep quality dimensions (Subjective sleep quality, Sleep latency, Sleep duration, Habitual sleep efficiency, Use of sleep medication, and Daytime dysfunction), all of which are weighted equally on a 0–3 scale to assess sleep quality over the previous month [43]. The seven component ratings are then combined to get a total PSQI sleep quality score ranging from 0 to 21, with 0 being the best and 21 representing the worst. A higher score indicates poorer sleep quality. The total PSQI global score correlation coefficient for test-retest reliability in primary insomnia patients was .87. As a marker for sleep disruptions in insomnia patients vs controls, a PSQI global score of 5 had a sensitivity of 98.7 and a specificity of 84.4 [44] The validated Bangla version of the PSQI, also known as the Bengali Pittsburgh sleep quality index BPSQI, was utilized to measure participants' sleep quality in the current study. A BPSQI score of 5 or above indicated poor sleep quality [45].

## Statistical analysis

SPSS version 23.0 (IBM Corp) and JASP statistical software were used for data analysis. Aside from descriptive statistics, other inferential maneuvers, such as Chi square tests, confirmatory factor analysis, reliability analysis, and binary logistic regression, were carried out as needed. Variables identified as having a significant connection in univariate analysis were included in the multivariate model adjusting for confounders. These variables included age, sex, income, place of residence, housing status, type of family, physical activity, duration/cost of internet use, time spent online, personal, social, academic, and romantic relationships, internet addiction, depression, perceived stress, anxiety, sleep quality, etc. Facebook Addiction served as the dependent variable in the analysis, and the sociodemographic and psychological variables served as the independent variables. The final results were presented with Wald statistics, level of significance, odds ratios, and 95% confidence intervals for odds ratios for each of the predictors.

## Result

Among the 720 participants in the current study, 55.4% were females (n = 399), and 44.6% were males (n = 321). On average, the respondents were around 21 years old. Eighty-three percent (n = 599) of the students were Muslims, while 15% (n = 111) were Hindus. According to their parents' occupations, their fathers were 30.7% (n = 221) businessmen and their mothers were 70% (n = 509) housewives. Seven percent of the medical students in the study sample (n = 51) were married, and three-quarters (n = 543) were self-employed. According to data, medical students lived in medical hostels 64% of the time (n = 462) and came from urban areas almost as frequently (n = 457). More than 20% of students experienced a deprived financial status (n = 166) and strained family connections (n = 157). Poor academic performance was reported by over one-third of those polled (n = 264), and 16.3% (n = 117) claimed they had negative ties with friends and family [Table 3].

According to the data, 63.7% of respondents (n = 469) had mild to severe anxiety, while 85.9% of medical students (n = 618) had moderate to high perceived stress. They had been

**Table 3. Social and individual characteristics related to Facebook addiction (N = 720).**

| Individual and social characteristics | | Bergen Facebook Addiction Scale BFAS's sub-domains | | | | | | | | | | | | Facebook Addiction BFAS Polythetic cut-off | | Facebook Addiction BFAS monothetic cut-off | |
|---|---|---|---|---|---|---|---|---|---|---|---|---|---|---|---|---|---|
| | | Salience | | Tolerance | | Mood modification | | Relapse | | Withdrawal | | Conflict | | | | | |
| | | No n (%) | Yes n (%) | No n (%) | Yes n (%) | No n (%) | Yes n (%) | No n (%) | Yes n (%) | No n (%) | Yes n (%) | No n (%) | Yes n (%) | No n (%) | Yes n (%) | No n (%) | Yes n (%) |
| **Financial situation** | Average [72.1%] | 339 (65.3) | 180 (34.7) | 209 (40.3) | 310 (59.7) | 362 (69.7) | 157 (30.3) | 347 (66.9) | 172 (33.1) | 401 (77.3) | 118 (22.7) | 307 (59.2) | 212 (40.8) | 390 (75.1) | 129 (24.9) | 500 (96.3) | 19 (3.7) |
| | Negative [23.0%] | 79 (47.6) | 87 (52.4) | 51 (30.7) | 115 (69.3) | 83 (50.0) | 83 (50.0) | 84 (50.6) | 82 (49.4) | 99 (59.6) | 67 (40.4) | 63 (38.0) | 103 (62.0) | 90 (54.2) | 76 (45.8) | 141 (84.9) | 25 (15.1) |
| | Positive [4.9%] | 24 (68.6) | 11 (31.4) | 13 (37.1) | 22 (62.9) | 29 (82.9) | 6 (17.1) | 19 (54.3) | 16 (45.7) | 26 (74.3) | 9 (25.7) | 14 (40) | 21 (60) | 28 (80) | 7 (20) | 33 (94.3) | 2 (5.70) |
| | Significance | $\chi^2$(2, N = 720) = 17.47, p = .000 | | $\chi^2$(2, N = 720) = 4.87, p = .08 | | $\chi^2$(2, N = 720) = 26.55, p = .000 | | $\chi^2$(2, N = 720) = 15.24, p = .000 | | $\chi^2$(2, N = 720) = 19.88, p = .000 | | $\chi^2$(2, N = 720) = 25.34, p = .000 | | $\chi^2$(2, N = 720) = 28.09, p = .000 | | $\chi^2$(2, N = 720) = 27.36, p = .000 | |
| **Academic performance** | Average [38.5%] | 202 (72.9) | 75 (27.1) | 134 (48.4) | 143 (51.6) | 203 (73.3) | 74 (26.7) | 205 (74) | 72 (26) | 226 (81.6) | 51 (18.4) | 198 (71.5) | 79 (28.5) | 231 (83.4) | 46 (16.6) | 269 (97.1) | 8 (2.9) |
| | Negative [36.6%] | 140 (53.0) | 124 (47.0) | 68 (25.8) | 196 (74.2) | 145 (54.9) | 119 (45.1) | 120 (45.5) | 144 (54.5) | 160 (60.6) | 104 (39.4) | 78 (29.5) | 186 (70.5) | 143 (54.2) | 121 (45.8) | 234 (88.6) | 30 (11.4) |
| | Positive [24.9%] | 100 (55.9) | 79 (44.1) | 71 (39.7) | 108 (60.3) | 126 (70.4) | 53 (29.6) | 125 (69.8) | 54 (30.2) | 140 (78.2) | 39 (21.8) | 108 (60.3) | 71 (39.7) | 134 (74.9) | 45 (25.1) | 171 (95.5) | 8 (4.5) |
| | Significance | $\chi^2$(2, N = 720) = 26.64, p = .000 | | $\chi^2$(2, N = 720) = 29.68, p = .00 | | $\chi^2$(2, N = 720) = 22.46, p = .000 | | $\chi^2$(2, N = 720) = 52.48, p = .000 | | $\chi^2$(2, N = 720) = 33.45, p = .000 | | $\chi^2$(2, N = 720) = 100.20, p = .000 | | $\chi^2$(2, N = 720) = 57.70, p = .000 | | $\chi^2$(2, N = 720) = 17.70, p = .000 | |
| **Friends and social relationship** | Average [48.7%] | 223 (63.5) | 128 (36.5) | 150 (42.7) | 201 (57.3) | 240 (68.4) | 111 (31.6) | 238 (67.8) | 113 (32.2) | 271 (77.2) | 80 (22.8) | 207 (59) | 144 (41) | 266 (75.8) | 85 (24.2) | 339 (96.6) | 12 (3.4) |
| | Negative [16.3%] | 63 (53.8) | 54 (46.2) | 29 (24.8) | 88 (75.2) | 59 (50.4) | 58 (49.6) | 50 (42.7) | 67 (57.3) | 65 (55.6) | 52 (44.4) | 39 (33.3) | 78 (66.7) | 64 (54.7) | 53 (45.3) | 93 (79.5) | 24 (20.5) |
| | Positive [35%] | 156 (61.9) | 96 (38.1) | 94 (37.3) | 158 (62.7) | 175 (69.4) | 77 (30.6) | 162 (64.3) | 90 (35.7) | 190 (75.4) | 62 (24.6) | 138 (54.8) | 114 (45.2) | 178 (70.6) | 74 (29.4) | 242 (96) | 10 (4) |
| | Significance | $\chi^2$(2, N = 720) = 3.51, p = .17 | | $\chi^2$(2, N = 720) = 12.07, p = .002 | | $\chi^2$(2, N = 720) = 14.816, p = .001 | | $\chi^2$(2, N = 720) = 24.06, p = .000 | | $\chi^2$(2, N = 720) = 21.98, p = .000 | | $\chi^2$(2, N = 720) = 23.50, p = .000 | | $\chi^2$(2, N = 720) = 18.78, p = .000 | | $\chi^2$(2, N = 720) = 46.67, p = .000 | |
| **Family relationship** | Average [64.9%] | 299 (64) | 168 (36) | 179 (38.3) | 288 (61.7) | 321 (68.7) | 146 (31.3) | 313 (67) | 154 (33) | 361 (77.3) | 106 (22.7) | 268 (57.4) | 199 (42.6) | 350 (74.9) | 117 (25.1) | 450 (96.4) | 17 (3.6) |
| | Negative [21.8%] | 82 (52.2) | 75 (47.8) | 47 (29.9) | 110 (70.1) | 82 (52.2) | 75 (47.8) | 66 (42) | 91 (58) | 77 (49) | 80 (51) | 60 (38.2) | 97 (61.8) | 79 (50.3) | 78 (49.7) | 13 (83.4) | 26 (16.6) |
| | Positive [13.3%] | 61 (63.5) | 35 (36.5) | 47 (49) | 49 (51) | 71 (74) | 25 (26) | 71 (74) | 25 (26) | 88 (91.7) | 8 (8.3) | 56 (58.3) | 40 (41.7) | 79 (82.3) | 17 (17.7) | 93 (96.9) | 3 (3.1) |
| | Significance | $\chi^2$(2, N = 720) = 7.11, p = .029 | | $\chi^2$(2, N = 720) = 9.25, p = .010 | | $\chi^2$(2, N = 720) = 17.48, p = .000 | | $\chi^2$(2, N = 720) = 37.50, p = .000 | | $\chi^2$(2, N = 720) =, p = .000 | | $\chi^2$(2, N = 720) = 18.46, p = .000 | | $\chi^2$(2, N = 720) = 41.65, p = .000 | | $\chi^2$(2, N = 720) = 34.77, p = .000 | |
| **Romantic relationship** | Average [79.7%] | 363 (59.8) | 211 (36.8) | 217 (37.8) | 357 (62.2) | 389 (67.8) | 185 (32.2) | 369 (64.3) | 205 (35.7) | 436 (76) | 138 (24) | 316 (55.1) | 258 (44.9) | 418 (72.8) | 156 (27.2) | 545 (94.9) | 29 (5.1) |
| | Negative [9.2%] | 33 (50) | 33 (50) | 24 (36.4) | 42 (63.6) | 33 (50) | 33 (50) | 32 (48.5) | 34 (51.5) | 33 (50) | 33 (50) | 27 (40.9) | 39 (59.1) | 32 (48.5) | 34 (51.5) | 52 (78.8) | 14 (21.2) |
| | Positive [11.1%] | 46 (57.5) | 34 (42.5) | 32 (40) | 48 (60) | 52 (65) | 28 (35) | 49 (61.3) | 31 (38.7) | 57 (71.3) | 23 (28.7) | 41 (51.2) | 39 (48.8) | 58 (72.5) | 22 (27.5) | 77 (96.3) | 3 (3.8) |
| | Significance | $\chi^2$(2, N = 720) = 4.95, p = .08 | | $\chi^2$(2, N = 720) = .21, p = .897 | | $\chi^2$(2, N = 720) = 8.34, p = .015 | | $\chi^2$(2, N = 720) = 6.37, p = .04 | | $\chi^2$(2, N = 720) = 20.41, p = .054 | | $\chi^2$(2, N = 720) = 4.91, p = .086 | | $\chi^2$(2, N = 720) = 17.04, p = .000 | | $\chi^2$(2, N = 720) = 26.89, p = .000 | |

using the social media site for an average of about five years; mean (SD) = 4.59 (1.86), and 86% (n = 620) reported they had at least one fake Facebook ID. Facebook addiction was observed in 29.4% (n = 212) of medical students using BFAS's polythetic scheme cut-off [32], (recommended for epidemiological screening), and in 6.4 percent (n = 46) of medical students using the monothetic classification of Facebook Addiction (recommended for clinical use) [32]. Using Chen's Internet addiction score cut-off of = >64 [30], 30.8 percent (n = 222) of the sample population were found to be addicted to the Internet. Data revealed, that 86.5 percent of respondents (n = 623) had poor sleep quality, and 29.3 percent (n = 201) had moderate to severe depressive symptoms [Table 4].

To ascertain the significance of the association between Facebook Addiction and psychosocial characteristics of the medical students, multiple chi-square tests were reported as chi-square statistics with sample size, degree of freedom, and p-value. The majority of the six BFAS subdomains (Salience, Tolerance, Mood modification, Relapse, Withdrawal, Conflict) as well as those based on the BFAS polythetic and monothetic schemes, were statistically significantly related with financial, academic, and relationship (family, friends, social, and romantic) status. Nevertheless, the respondent's financial status was unaffected by the BFAS's "Tolerance" subdomain. The respondents' "Salience" and "conflict" responses revealed no associations between them and their social or romantic relationships [Table 3].

In a bivariate analysis of the study data, all levels of anxiety, perceived stress, depression, Internet Addiction, and sleep quality were statistically significantly associated with all six domains of the Bergen Facebook Addiction Scale [61]. Table demonstrated a significant relationship between Facebook Addiction and the respondents' behavioral traits using both the BFAS polythetic cut-off and the monothetic criteria [Table 4].

## The social and psychological predictors of Facebook addiction

In this study, no particular behavioral model was used to evaluate the determinants of Facebook addiction. Binary logistic regression employing the "enter" strategy was used to explore the effects of a variety of demographic, psychological, and behavioral traits on the chance that respondents had a Facebook addiction. The multivariate model included 19 independent variables that had previously demonstrated statistically significant relationship in bivariate analysis (sex, personal income, marital status, residence, place of living, family status, since when start using Facebook, expenditure related to Facebook use, relationship with friends and society, family relationship, romantic relationship, academic performance, quit attempt to excessive Facebook use, sports and exercise, anxiety, perceived stress, depression, Internet Addiction, and sleep quality).

The model was preliminarily checked for sampling adequacy and multicolinearity. According to a study, in observational studies with large populations that employ logistic regression in the analysis, a minimum sample size of 500 must be taken to produce statistics that correctly reflect the parameters [46]. We assessed multicollinearity by examining correlation indices and variance inflation factor (VIF) values. The magnitudes of the calculated coefficients of the predictor variables were less than .80, indicating that they were not highly correlated. The VIF values of the predictor variables, however, were likewise observed to be below 5.00.

The full model containing all predictors was statistically significant, $\chi2$ (41, N = 720) = 271.86, p < .001, indicating that the model was able to distinguish between respondents who reported and did not report a Facebook Addiction. The model as a whole explained between 31.5% (Cox and Snell R square) and 45% (Nagelkerke R squared) of the variance in Facebook Addiction, and correctly classified 80.7 percent of cases. As shown in Table 5, nine predictors made a statistically significant contribution (personal income, living with parents and family,

Table 4. Psychological and behavioral characteristics and Facebook addiction (N = 720).

| Psychological characteristics | | Bergen Facebook Addiction Scale BFAS's sub-domains | | | | | | | | | | | | Facebook Addiction BFAS Polythetic cut-off | | Facebook Addiction BFAS monothetic cut-off | |
|---|---|---|---|---|---|---|---|---|---|---|---|---|---|---|---|---|---|
| | | Salience | | Tolerance | | Mood modification | | Relapse | | Withdrawal | | Conflict | | | | | |
| | | No n (%) | Yes n (%) | No n (%) | Yes n (%) | No n (%) | Yes n (%) | No n (%) | Yes n (%) | No n (%) | Yes n (%) | No n (%) | Yes n (%) | No n (%) | Yes n (%) | No n (%) | Yes n (%) |
| Anxiety | Minimal [36.3%] | 192 (73.6) | 69 (26.4) | 121 (46.4) | 140 (53.6) | 194 (74.3) | 67 (25.7) | 200 (76.6) | 61 (23.4) | 221 (84.7) | 40 (15.3) | 180 (69) | 81 (31) | 224 (85.8) | 37 (14.2) | 258 (98.9) | 3 (1.1) |
| | Mild [37.2%] | 167 (62.3) | 101 (37.7) | 110 (41) | 158 (59) | 183 (68.3) | 85 (31.7) | 166 (61.9) | 102 (38.1) | 184 (68.7) | 84 (31.3) | 137 (51.1) | 131 (48.9) | 194 (72.4) | 74 (27.6) | 252 (94) | 16 (6) |
| | Moderate [19.4%] | 61 (43.6) | 79 (56.4) | 25 (17.9) | 115 (82.1) | 76 (54.3) | 64 (45.7) | 58 (41.4) | 82 (58.6) | 89 (63.6) | 51 (36.4) | 46 (32.9) | 94 (67.1) | 63 (45) | 77 (55) | 122 (87.1) | 18 (12.9) |
| | Severe [7.1%] | 22 (43.1) | 29 (56.9) | 17 (33.3) | 34 (66.7) | 21 (41.2) | 30 (58.8) | 26 (51) | 25 (49) | 32 (67.7) | 19 (37.3) | 21 (41.2) | 30 (58.8) | 27 (52.9) | 24 (47.1) | 42 (82.4) | 9 (17.6) |
| | Significance | $x^2$(3, N = 720) = 42.33, p = .000 | | $x^2$(3, N = 720) = 33.41, p = .000 | | $x^2$(3, N = 720) = 31.18, p = .000 | | $x^2$(3, N = 720) = 51.67, p = .000 | | $x^2$(3, N = 720) = 29.68, p = .000 | | $x^2$(3, N = 720) = 52.76, p = .000 | | $x^2$(3, N = 720) = 81.35, p = .000 | | $x^2$(3, N = 720) = 32.66, p = .000 | |
| Perceived stress | Low [14.2%] | 81 (79.4) | 21 (20.6) | 61 (59.8) | 41 (40.2) | 84 (82.4) | 18 (17.6) | 89 (87.3) | 13 (12.7) | 99 (97.1) | 3 (2.9) | 70 (77.5) | 23 (22.5) | 97 (95.1) | 5 (4.9) | 101 (99) | 1 (1) |
| | Moderate [75.3%] | 329 (60.7) | 213 (39.3) | 200 (36.9) | 342 (63.1) | 357 (65.9) | 185 (34.1) | 331 (61.1) | 211 (38.9) | 383 (70.7) | 159 (29.3) | 280 (51.7) | 262 (48.3) | 377 (69.6) | 165 (30.4) | 510 (94.1) | 32 (5.9) |
| | High [10.6%] | 32 (42.1) | 44 (57.9) | 12 (15.8) | 64 (84.2) | 33 (43.4) | 43 (56.6) | 30 (39.5) | 50 (60.5) | 44 (57.9) | 32 (42.1) | 25 (32.9) | 51 (67.1) | 34 (44.7) | 42 (55.3) | 63 (82.9) | 13 (17.1) |
| | Significance | $x^2$(2, N = 720) = 26.01, p = .000 | | $x^2$(2, N = 720) = 36.80, p = .000 | | $x^2$(2, N = 720) = 29.35, p = .000 | | $x^2$(2, N = 720) = 44.34, p = .000 | | $x^2$(2, N = 720) = 40.30, p = .000 | | $x^2$(2, N = 720) = 37.20, p = .000 | | $x^2$(2, N = 720) = 46.42, p = .000 | | $x^2$(2, N = 720) = 19.79, p = .000 | |
| Depression | No [30.8%] | 155 (69.8) | 67 (30.2) | 111 (50) | 111 (50) | 178 (80.2) | 44 (19.8) | 171 (77) | 51 (23) | 196 (88.3) | 26 (11.7) | 154 (69.4) | 68 (30.6) | 194 (87.4) | 28 (12.6) | 219 (98.6) | 3 (1.4) |
| | Mild [39.9%] | 187 (65.2) | 100 (34.8) | 106 (36.9) | 181 (63.1) | 192 (66.9) | 95 (33.1) | 180 (62.7) | 107 (37.3) | 205 (71.4) | 82 (28.6) | 146 (50.9) | 141 (49.1) | 203 (70.7) | 84 (29.3) | 272 (94.8) | 15 (5.2) |
| | Moderate [19.9%] | 75 (52.4) | 68 (47.6) | 47 (32.9) | 96 (67.1) | 74 (51.7) | 69 (48.3) | 78 (54.5) | 65 (45.5) | 88 (61.5) | 55 (38.5) | 67 (46.9) | 76 (53.1) | 88 (61.5) | 55 (38.5) | 132 (92.3) | 11 (7.7) |
| | Moderately severe [7.5%] | 20 (37) | 34 (63) | 7 (13) | 47 (87) | 25 (46.3) | 29 (53.7) | 17 (31.5) | 37 (68.5) | 33 (61.1) | 21 (38.9) | 14 (25.9) | 40 (74.1) | 19 (35.2) | 35 (64.8) | 43 (79.6) | 11 (20.4) |
| | Severe [1.9%] | 5 (35.7) | 9 (64.3) | 2 (14.3) | 12 (85.7) | 5 (35.7) | 9 (64.3) | 4 (28.6) | 10 (71.4) | 4 (28.6) | 10 (71.4) | 3 (21.4) | 11 (78.6) | 4 (28.6) | 10 (71.4) | 8 (57.1) | 6 (42.9) |
| | Significance | $x^2$(4, N = 720) = 30.60, p = .000 | | $x^2$(4, N = 720) = 33.04, p = .000 | | $x^2$(4, N = 720) = 47.88, p = .000 | | $x^2$(4, N = 720) = 52.90, p = .000 | | $x^2$(4, N = 720) = 54.18, p = .000 | | $x^2$(4, N = 720) = 48.07, p = .000 | | $x^2$(4, N = 720) = 80.27, p = .000 | | $x^2$(4, N = 720) = 59.26, p = .000 | |
| Internet Addiction | No [69.2%] | 350 (70.3) | 148 (29.7) | 232 (46.6) | 266 (53.4) | 368 (73.9) | 130 (26.1) | 375 (75.3) | 123 (24.7) | 415 (83.3) | 83 (16.7) | 332 (66.7) | 166 (33.3) | 422 (84.7) | 76 (15.3) | 493 (99) | 5 (1) |
| | Yes [30.8%] | 92 (41.4) | 130 (58.6) | 41 (18.5) | 181 (81.5) | 106 (47.7) | 116 (52.3) | 75 (33.8) | 147 (66.2) | 111 (50) | 111 (50) | 52 (23.4) | 170 (76.6) | 86 (38.7) | 136 (61.3) | 181 (815) | 41 (18.5) |
| | Significance | $x^2$(1, N = 720) = 53.68, p = .000 | | $x^2$(4, N = 720) = 51.57, p = .000 | | $x^2$(4, N = 720) =, 46.67 p = .000 | | $x^2$(4, N = 720) = 112.93, p = .000 | | $x^2$(4, N = 720) = 86.67, p = .000 | | $x^2$(4, N = 720) = 115.37, p = .000 | | $x^2$(4, N = 720) = 156.4, p = .000 | | $x^2$(4, N = 720) = 78.31, p = .000 | |
| Sleep quality | Good sleep [13.5%] | 71 (73.2) | 26 (26.8) | 48 (49.5) | 49 (50.5) | 79 (81.4) | 18 (18.6) | 79 (81.4) | 18 (18.6) | 85 (87.6) | 12 (12.4) | 65 (67) | 32 (33) | 86 (88.7) | 11 (11.3) | 97 (100) | 0 (0) |
| | Poor sleep [86.5%] | 371 (59.6) | 252 (40.4) | 225 (36.1) | 398 (63.9) | 395 (63.4) | 228 (36.6) | 371 (59.6) | 252 (40.4) | 441 (70.8) | 182 (29.2) | 319 (51.2) | 304 (48.8) | 422 (67.7) | 201 (32.3) | 577 (92.6) | 46 (7.4) |
| | Significance | $x^2$(1, N = 720) = 6.59, p = .010 | | $x^2$(1, N = 720) = 6.37, p = .012 | | $x^2$(1, N = 720) = 12.14, p = .000 | | $x^2$(1, N = 720) = 17.16, p = .000 | | $x^2$(1, N = 720) = 12.01, p = .001 | | $x^2$(1, N = 720) = 8.43, p = .004 | | $x^2$(1, N = 720) = 17.69, p = .004 | | $x^2$(1, N = 720) = 7.65, p = .006 | |

**Table 5. Predictors for Facebook Addiction [Dependent variable: BFAS polythetic scheme-based Facebook Addiction categories [32]].**

| | Wald | Sig. | Exp(B) | 95% C.I.for EXP(B) Lower | 95% C.I.for EXP(B) Upper |
|---|---|---|---|---|---|
| No personal income | 5.961 | .015 | 1.825 | 1.126 | 2.959 |
| Living with parents & family | 4.250 | .039 | .367 | .142 | .952 |
| Positive family relationship | 5.348 | .021 | .398 | .182 | .869 |
| Poor academic performance | 11.284 | .001 | 2.457 | 1.454 | 4.150 |
| Internet Addiction | 62.861 | .000 | 6.073 | 3.888 | 9.486 |
| Moderate anxiety | 6.358 | .012 | 2.451 | 1.221 | 4.921 |
| Moderately severe depression | 3.619 | .054 | 2.623 | .971 | 7.084 |
| Moderate perceived stress | 9.617 | .002 | 5.867 | 1.918 | 17.949 |
| High perceived stress | 3.734 | .053 | 3.646 | .982 | 13.543 |

The table shows Wald statistics, odds ratios, and 95% confidence intervals for odds ratios for each of the predictors.

positive family relations, poor academic performance, Internet addiction, moderate and severe anxiety, moderately severe depression, moderate and high perceived stress). Medical students with no personal income were nearly twice as likely as those with personal income to be addicted to Facebook [OR = 1.825, (1.126, 2.959), p = .015]. Respondents, who lived with their parents, were 2.7 times less likely to be addicted to Facebook than those who lived in a hostel [OR with 95 percent CI. 367, (.142, .952), p = .039] [Table 5].

Medical students who perceived positive family relationships were 2.5 times less likely to develop Facebook Addiction than those who did not [OR with 95 percent CI, .398 (.182, .869), p = .021].Respondents with poor academic performance were about 2.5 times more prone to develop Facebook Addiction [OR with 95 percent CI, 2.457 (1.454, 4.15), p = .001].Medical students who experienced moderate anxiety symptoms, as well as those who experienced moderately severe depression, were about 2.5 times more likely to have a Facebook addiction. The strongest predictor of Facebook addiction was found to be Internet Addiction [OR with 95 percent CI, 6.073 (3.88, 9.49), p = .000], followed by moderate perceived stress [OR with 95 percent CI, 5.867 (1.918, 17.949), p = .002]. When compared to students who did not experience any stress symptoms, medical students who reported high perceived stress were 3.65 times more likely to have a Facebook addiction. [OR with 95 percent CI, 3.65 (.982, 13.54), p = .053] [Table 5].

## Discussion

The purpose of this cross-sectional study was to determine the prevalence of Facebook Addiction among medical students, as well as its potential psychosocial and behavioral predictors.

The prevalence of Facebook addiction among medical students was found to be 29.4% (n = 212) in our study using the BFAS's polythetic scheme cut-off (e.g., scoring three or higher on at least four of the six items), which is recommended for epidemiological screening. Almost identical findings were observed in studies conducted in India [47] and Nepal [48]. However, using the same criteria to identify FA among college and university students, several other Bangladeshi studies discovered an eight to eleven percent higher prevalence of FA 36.9% [26], 39.7% [7].

According to our findings, 63.7 percent of respondents (n = 469) experienced mild to severe anxiety, which is nearly double the result of a meta-analysis of 69 studies involving 40,348 medical students, which found that the overall prevalence of anxiety was 33.8% [49], but is consistent with the findings of similar studies conducted in India [50] and Egypt [51].

According to the data, moderately anxious medical students were more than twice as likely to be addicted to Facebook, and similar findings were seen among Pakistani students [22]. However, addiction to Facebook and severe anxiety did not appear to be significantly associated in this study.It is interesting to note that because Facebook is a social network, addiction to it might have a stronger link to social anxiety than general anxiety. One of the possible explanations is that people with severe social anxiety, also known as social phobia, may avoid all social interactions because they feel uneasy and anxious when they engage in activities that other people find "normal."

The current study used PSS10B to assess perceived stress, and 85.9% of the medical students reported moderate to severe stress. However, a comparative study of public and private medical students in Bangladesh used GHQ-12 to assess stress, and the overall prevalence of stress was found to be 54% [52]. Because it is more about their feelings about lack of control and unpredictability than actual stressors, the current prevalence of perceived stress may be higher than actual stress. This study found that Facebook users who experienced moderate to high levels of stress are about four to six times more likely to develop Facebook addictions, which is consistent with earlier findings [17,20].

In our study, 30.8 percent (n = 222) of medical students had internet addiction, which is slightly higher than in two earlier studies in Bangladesh [53,54] but about ten percent lower than that of other Middle Eastern nations like Jordan (40%) [55] and Iran (39.6%) [56]. Depression was observed in 29.3% (n = 201) of medical students in our study, which is five times higher than the national prevalence of depression among adults [57]. Almost alike prevalence had also been observed among medical students in Thailand [58] and Nepal [59]. A meta-analysis of 77 studies that included 62,728 medical students and 1,845 non-medical students revealed a 28.0% global prevalence of depression among medical students [60].

In our study, medical students with moderately severe depression were 2.5 times more likely to develop a Facebook addiction. The results of the current study support the findings of previous studies that have shown links between FA and depression [7,18,22]. The American Academy of Pediatrics (AAP) coined the term "Facebook depression" to describe the condition of teenagers who were heavily using Facebook and displaying depressive symptoms [61]. It is noteworthy that no correlation between severe depression and Facebook addiction was found in our study. This could be a result of psychomotor retardation, one of the main symptoms of severe depression, which can prevent people from engaging in social interaction and increase their susceptibility to fatigue [62].

Our data showed that factors such as, perceived positive family relationships, poor academic performance, moderate anxiety, moderately severe depression, Internet addiction, and moderate to high perceived stress were statistically significantly associated with Facebook Addiction. Many other studies also cited these demographic, psychosocial, and behavioral factors as significant predictors of Facebook addiction [14–16,21–24,26,27]. Personal financial situation and living with family members were found to be significant determinants of FA among medical students, which had not previously been recorded in Bangladeshi research. Though it is obvious that medical students will remain in a dormitory, bi-annual parent meetings might be held to increase student-family relations and enhance the supporting communication triangle among student, family, and institution.

## Strength and limitations

The study was conducted on a randomly chosen group of medical students using appropriate and reliable research tools. Insight into the temporality of the relationship between variables, however, is lacking because the current study is cross-sectional. Another flaw is that medical

students' self-reported mental health outcomes were not compared to clinical diagnoses. The data collection method should also include a qualitative approach to explore concepts and revelations regarding the behavioral and psychometric responses of the study participants.

## Conclusion

Due to a number of modifiable psychosocial factors, a significant proportion of medical students were found to be addicted to Facebook. Regular screening of medical students for Facebook addiction and its related issues should be prioritized in order to identify the scale of Facebook addiction and to improve medical students' quality of life in terms of their mental health. Public health and psychiatry departments at respective medical schools should coordinate specific behavioral and psychological treatments aimed at reducing anxiety, stress, and depression, as well as strengthening family relationships that may be recommended to parents.

Students who are already addicted to Facebook may benefit from interventions based on the BASNEF consultation model (Beliefs, Attitudes, Subjective Norms, and Enabling Factors) and the commonly used counseling steps using GATHER-Greet, Ask, Tell Clients about Their Choices, Help, Explain, and Return for Follow-Up. More study is needed to assess the efficiency of such common cost-effective intervention strategies.

## Acknowledgments

The Directorate General of Health Services and the Bangladesh Medical Research Council have provided institutional and administrative support. We are also grateful to the administration of the Public Medical Colleges and students for their unwavering support and cooperation.

## Author Contributions

**Conceptualization:** Md Rizwanul Karim.

**Formal analysis:** Md Rizwanul Karim, Md. Jawadul Haque.

**Investigation:** Md Rizwanul Karim, Helal Uddin Ahmed.

**Methodology:** Md. Jawadul Haque, Helal Uddin Ahmed.

**Resources:** Shahnaz Akhter.

**Supervision:** Md. Jawadul Haque, Shahnaz Akhter.

**Validation:** Md Rizwanul Karim, Shahnaz Akhter, Helal Uddin Ahmed.

**Writing – original draft:** Md Rizwanul Karim.

**Writing – review & editing:** Shahnaz Akhter, Helal Uddin Ahmed.

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
