## [Decision Letter · Decision Letter 0]

30 Aug 2022

PGPH-D-22-01246

Facebook addiction and its related factors in medical students; Bangladesh context

Dear Dr. Karim,

Thank you for submitting your manuscript to PLOS Global Public Health. After careful consideration, we feel that it has merit but does not fully meet PLOS Global Public Health’s publication criteria as it currently stands. Therefore, we invite you to submit a revised version of the manuscript that addresses the points raised during the review process.

We look forward to receiving your revised manuscript.

Kind regards,

Muhammad Fawad Rasool

Academic Editor

Journal Requirements:

a. Please clarify all sources of funding (financial or material support) for your study. List the grants (with grant number) or organizations (with url) that supported your study, including funding received from your institution. 

b. State the initials, alongside each funding source, of each author to receive each grant.

c. State what role the funders took in the study. If the funders had no role in your study, please state: “The funders had no role in study design, data collection and analysis, decision to publish, or preparation of the manuscript.”

2. Please note that your Data Availability Statement is currently missing [the repository name. If your manuscript is accepted for publication, you will be asked to provide these details on a very short timeline. We therefore suggest that you provide this information now, though we will not hold up the peer review process if you are unable.

3. We do not publish any copyright or trademark symbols that usually accompany proprietary names, eg (R), (C), or TM  (e.g. next to drug or reagent names). Please remove all instances of trademark/copyright symbols throughout the text, including © on line 500.

4. Please provide separate figure files in .tif or .eps format and remove the embedded figures from the manuscript file.

Additional Editor Comments (if provided):

Reviewers' comments:

Reviewer's Responses to Questions

**Comments to the Author**

1. Does this manuscript meet PLOS Global Public Health’s publication criteria? Is the manuscript technically sound, and do the data support the conclusions? The manuscript must describe methodologically and ethically rigorous research with conclusions that are appropriately drawn based on the data presented.

Reviewer #1: Partly

Reviewer #2: Yes

2. Has the statistical analysis been performed appropriately and rigorously?

Reviewer #1: Yes

Reviewer #2: Yes

3. Have the authors made all data underlying the findings in their manuscript fully available (please refer to the Data Availability Statement at the start of the manuscript PDF file)?

Reviewer #1: No

Reviewer #2: No

4. Is the manuscript presented in an intelligible fashion and written in standard English?

Reviewer #1: No

Reviewer #2: Yes

5. Review Comments to the Author

Reviewer #1: First of all, the title of the study should include the study design which is a "cross-sectional survey". Second, the abstract should comprise the main statistical tests that have been used to identify the predictors(i.e. OR). Third, I would rather that the study follows a certain model to justify the section of the potential predictors. For instance, Anderson Behavioral Model. Furthermore, the study has not identified whether the respondents had consented and what type of consent was used ( written or verbal). Moreover, the study was not clearly identified the language used in data collection (i.e. English or any other language).

There were nineteen independent variables that entered the analysis using Binary Logistic Regression (BLR), however, the study had not provided a robust justification for compromising only 19 out of the rest of the independent variables.

The discussion part of the study did not make clear what this research adds to existing research. Furthermore, the study did not describe how this research advances our understanding or how it may inspire future applications. For instance, the study declared that "Our data showed that factors such as personal income, living with their family, perceived positive, family relationships, poor academic performance, moderate anxiety, moderately severe, depression, Internet addiction, and moderate to high perceived stress were statistically, significantly associated with Facebook addiction", however, it had not provided a possible interpretation.

The implications of the study should specify the type of intervention that is recommended to tackle the problem of Facebook addiction(e.g. an educational or behavioral intervention) and the target population based on the risk factors identified in this study.

Reviewer #2: well planned executed and presented study.

intorduction - impact of overall social media on medical students can be added.

need to add more information about questionnaires used. validation, reliability and scoring need to be added.

6. PLOS authors have the option to publish the peer review history of their article (what does this mean?). If published, this will include your full peer review and any attached files.

**Do you want your identity to be public for this peer review?** For information about this choice, including consent withdrawal, please see our Privacy Policy.

Reviewer #1: **Yes: **Khalid Awad Al Kubaissi

Reviewer #2: No

---

## [Decision Letter · Decision Letter 1]

5 Dec 2022

PGPH-D-22-01246R1

Facebook Addiction and its related factors among medical students; a cross- sectional study in Bangladesh

Dear Dr. Karim,

Thank you for submitting your manuscript to PLOS Global Public Health. After careful consideration, we feel that it has merit but does not fully meet PLOS Global Public Health’s publication criteria as it currently stands. Therefore, we invite you to submit a revised version of the manuscript that addresses the points raised during the review process.

EDITOR: Please insert comments here and delete this placeholder text when finished. Be sure to:

Indicate which changes you require for acceptance versus which changes you recommendAddress any conflicts between the reviews so that it's clear which advice the authors should followProvide specific feedback from your evaluation of the manuscript

Please ensure that your decision is justified on PLOS Global Public Health’s publication criteria and not, for example, on novelty or perceived impact.

We look forward to receiving your revised manuscript.

Kind regards,

Muhammad Fawad Rasool

Academic Editor

Journal Requirements:

Additional Editor Comments (if provided):

Reviewers' comments:

Reviewer's Responses to Questions

**Comments to the Author**

1. If the authors have adequately addressed your comments raised in a previous round of review and you feel that this manuscript is now acceptable for publication, you may indicate that here to bypass the “Comments to the Author” section, enter your conflict of interest statement in the “Confidential to Editor” section, and submit your "Accept" recommendation.

Reviewer #1: (No Response)

Reviewer #2: All comments have been addressed

2. Does this manuscript meet PLOS Global Public Health’s publication criteria? Is the manuscript technically sound, and do the data support the conclusions? The manuscript must describe methodologically and ethically rigorous research with conclusions that are appropriately drawn based on the data presented.

Reviewer #1: Partly

Reviewer #2: Yes

3. Has the statistical analysis been performed appropriately and rigorously?

Reviewer #1: Yes

Reviewer #2: Yes

4. Have the authors made all data underlying the findings in their manuscript fully available (please refer to the Data Availability Statement at the start of the manuscript PDF file)?

Reviewer #1: Yes

Reviewer #2: Yes

5. Is the manuscript presented in an intelligible fashion and written in standard English?

Reviewer #1: No

Reviewer #2: Yes

6. Review Comments to the Author

Reviewer #1: Please check the following points:

Prior to running the analysis for the BLR, important underlying assumptions should be checked such as sample size and multicollinearity.

Please specify the time of the discussion session for the participants

Please specify the research team

Please provided the consent form and specify its type(verbal, written...)

Please specify how participant privacy and confidentiality were assured?

The section of predicators variables was based on the literature review, please sure that you provided the appropriate citations in every section of the manuscript.

Please specify if all study variables were entered into the BLR model (All entered) or any other approach

Please make the appropriate editing and grammars checking for all the manuscript

Reviewer #2: (No Response)

7. PLOS authors have the option to publish the peer review history of their article (what does this mean?). If published, this will include your full peer review and any attached files.

**Do you want your identity to be public for this peer review?** For information about this choice, including consent withdrawal, please see our Privacy Policy.

Reviewer #1: No

Reviewer #2: No

---

## [Editor Report · Decision Letter 2]

24 Jan 2023

Facebook Addiction and its related factors among medical students; a cross- sectional study in Bangladesh

PGPH-D-22-01246R2

Dear Dr Karim,

We are pleased to inform you that your manuscript 'Facebook Addiction and its related factors among medical students; a cross- sectional study in Bangladesh' has been provisionally accepted for publication in PLOS Global Public Health.

Best regards,

Muhammad Fawad Rasool

Academic Editor

Reviewer Comments (if any, and for reference):

<quillbot-extension-portal></quillbot-extension-portal>